# QUANTIZED SPARSE PCA FOR NEURAL NETWORK WEIGHT COMPRESSION

## ABSTRACT

In this paper, we introduce a novel method of weight compression. In our method, we store weight tensors as sparse, quantized matrix factors, whose product is computed on the fly during inference to generate the target model's weight tensors. The underlying matrix factorization problem can be considered as a quantized sparse PCA problem and solved through iterative projected gradient descent methods. Seen as a unification of weight SVD, vector quantization and sparse PCA, our method achieves or is on par with state-of-the-art trade-offs between accuracy and model size. Our method is applicable to both moderate compression regime, unlike vector quantization, and extreme compression regime.

## 1 INTRODUCTION

Deep neural networks have achieved state-of-the-art results in a wide variety of tasks. However, deployment remains challenging due to their large compute and memory requirements. Neural networks deployed on edge devices such as mobile or IoT devices are subject to stringent compute and memory constraints, while networks deployed in the cloud do not suffer such constraints but might suffer excessive latency or power consumption.

To reduce neural network memory and compute footprint, several approaches have been introduced in literature. Methods related to our approach, as well as their benefits and downsides, are briefly introduced in this section and described in more detail in the related works section. *Tensor factorization* approaches (Denil et al., 2013) replace a layer in a neural network with two layers, whose weights are low-rank factors of the original layer's weight tensor. This reduces the number of parameters and multiply-add operations (MACs), but since the factorizations are by design restricted to those that can be realized as individual layers, potential for compression is limited. By *pruning* a neural network (Louizos et al., 2017; He et al., 2017), individual weights are removed. Pruning has shown to yield moderate compression-accuracy trade-offs. Due to the overhead required to keep track of which elements are pruned, real yield of (unstructured) pruning is lower than the pruning ratio. An exception to this is *structured pruning*, in which entire neurons are removed from a network, and weight tensors can be adjusted accordingly. However, achieving good compression ratios at reasonable accuracy using structured pruning has proven difficult. *Scalar quantization* (Jacob et al., 2018; Nagel et al., 2021) approximates neural network weights with fixed-point values, i.e., integer values scaled by a fixed floating point scalar. Scalar quantization has shown to yield high accuracy at reasonable compression ratios, e.g., 8 bit quantization yields a 4x compression ratio and virtually no accuracy degradation on many networks. However, scalar quantization does not yield competitive compression vs accuracy trade-offs at high compression ratios. *Vector quantization* (Stock et al., 2019; Martinez et al., 2021) approximates small subsets of weights, e.g., individual $3 \times 3$ filters in convolutional weight tensors, by a small set of codes. This way, the storage can be reduced by storing the codebook and one code index for each original vector, instead of the individual weights. While vector quantization can achieve high compression with moderate accuracy loss, these methods usually struggle to reach the accuracy of uncompressed models in low compression regimes.

In this paper, we provide a novel view on tensor factorization. Instead of restricting factorization to those that can be realized as two separate neural network layers, we show that much higher compression ratios can be achieved by shifting the order of operations. In our method, we find a factorization $\mathbf{C}, \mathbf{Z}$ for an original weight tensor $\mathbf{W}$, such that the matrix product $\mathbf{CZ}$ closely approximates the

Figure 1: An illustration of application of Quantized Sparse PCA to a convolutional weight tensor. The weight tensor is reshaped into a matrix of shape $d \times n$ and factorized into a codebook $\mathbf{C} \in \mathbb{R}^{d \times k}$ and a latent $\mathbf{Z} \in \mathbb{R}^{k \times n}$. Both factors are quantized while only $\mathbf{Z}$ is sparse. During inference the arrows are followed backwards: the reshaped and tiled matrix is computed from the product of the factors $\mathbf{C}$ and $\mathbf{Z}$. The result is reshaped into the original weight tensor.

original weight tensor. The compression is then pushed further by obtaining quantized factors and additionally sparse $\mathbf{Z}$. During inference, the product $\mathbf{CZ}$ is computed first, and its result is reshaped back into the original weight tensor's shape, and used for the following computations. This approach allows the use of an arbitrary factorization of the original weight tensor.

We show that this approach outperforms or is on par with vector quantization in high compression regimes, yet extends to scalar quantization levels of compression-accuracy trade-offs for lower compression ratios.

Our contributions in this paper are as follows:

- We show that the problem of tensor factorization and vector quantization can be formulated in a unified way as quantized sparse principle component analysis (PCA) problem.

- We propose an iterative projected gradient descent method to solve the quantized sparse PCA problem.

- Our experimental results demonstrate the benefits of this approach. By simultaneously solving tensor factorization and vector quantization problem, we can achieve better accuracy than vector quantization in low compression regimes, and higher compression ratios than scalar quantization approaches at moderate loss in accuracy.

## 2 RELATED WORK

**SVD-based methods and tensor decompositions** SVD decomposition was first used to demonstrate redundancy in weight parameters in neural networks in Denil et al. (2013). Later several methods for reducing the inference time based on SVD decomposition were suggested (Denton et al., 2014; Jaderberg et al., 2014). A similar technique was proposed for gradients compression in data-parallel distributed optimization by Vogels et al. (2019). The main difference between these methods is the way 4D weights of a convolutional layer is transformed into a matrix which leads to different shapes of the convolutional layers in the resulting decomposition. Following a similar direction, several works focus on higher-order tensor decomposition methods which lead to introducing of three or four convolutional layers (Lebedev et al., 2014; Kim et al., 2015; Su et al., 2018).

**Weight pruning** A straightforward approach to reducing neural network model size is removing a percentage of weights. A spectrum of weight pruning approaches of different granularity has been introduced in the literature. Structured pruning approaches such as He et al. (2017) kill entire channels of the weights, while unstructured pruning approaches (Louizos et al., 2017; Zhu & Gupta, 2017; Neklyudov et al., 2017; Dai et al., 2018) focus on individual values. A recent survey on unstructured pruning is provided in Gale et al. (2019).

**Scalar quantization and mixed precision training** By quantizing neural network weights to lower bitwidths, model footprint can be reduced as well, as each individual weight requires fewer

bits to be stored. For example, quantizing 32 bit floating points weight to 8 bit fixed point weights yields a 4x compression ratio. Most quantization approaches use the straight-through estimator (STE) for training quantized models (Bengio et al. (2013); Krishnamoorthi (2018)). One way to further improve the accuracy of quantized models is learning the quantization scale and offset jointly with the network parameters (Esser et al. (2019); Bhalgat et al. (2020)). A recent survey on practical quantization approaches can be found in Nagel et al. (2021).

In order to improve the accuracy of quantized models, several methods suggest using mixed precision quantization. The work by Uhlich et al. (2019) introduced an approach on learning integer bit-width for each layer using STE. Using non-uniform bit-width allows the quantization method to use lower bit-width for more compressible layers of the network. Several works (van Baalen et al., 2020; Dong et al., 2019; Wang et al., 2019) improve upon the approach by Uhlich et al. (2019) by using different methods for optimization over the bit-widths.

**Vector quantization.** Several works use vector quantization approach for compression of weights of convolutional and fully connected layers (Gong et al. (2014); Martinez et al. (2021); Fan et al. (2020); Stock et al. (2019); Wu et al. (2016)). The convolutional weight tensors are reshaped into matrices, then K-means methods is applied directly on the rows or columns. Besides weight compression, the work by Wu et al. (2016) suggests using vector quantization for reducing the inference time by reusing parts of the computation.

Recently several works suggested improvements on the basic vector quantization approach. Data-aware vector quantization which improves the clustering method by considering input activation data is demonstrated to improve the accuracy of the compressed models by Stock et al. (2019). Another direction is introducing a permutation to the weight matrices which allows to find subsets of weights which are more compressible (Martinez et al., 2021). An inverse permutation is applied at the output of the corresponding layers to preserve the original output of the model.

Besides weights compression problem, various improved vector quantization methods were applied in image retrieval domain in order to accelerate scalar product computation for image descriptors (Chen et al. (2010); Ge et al. (2013); Norouzi & Fleet (2013)). In particular, additive quantization method which we will show is related to quantized sparse PCA was introduced Babenko & Lempitsky (2014). Surveys on vector quantization methods are provided in Matsui et al. (2018); Gersho & Gray (2012).

**Sparse PCA.** Introduced in Zou et al. (2006), sparse PCA can be solved by a plethora of algorithms. The method proposed in this paper can be considered as an instance of thresholding algorithms (Ma, 2013). Although soft-thresholding methods are prevalent in the literature, we adopt an explicit projection step using hard thresholding to have direct control over the compression ration. Note that sparse PCA can be extended to include additional structures with sparsity (Jenatton et al., 2010; Lee et al., 2006).

## 3 METHOD

In this section, we describe the main algorithm, which can be considered as sparse quantized principle component analysis (PCA).

### 3.1 QUANTIZED SPARSE PCA

Consider a weight tensor of a convolutional layer $\mathbf{W} \in \mathbb{R}^{f_{out} \times f_{in} \times h \times w}$, where $f_{in}$, $f_{out}$ are the number of input and output feature maps, and $h$, $w$ are the spatial dimensions of the filter. We reshape it into a matrix $\widetilde{\mathbf{W}} \in \mathbb{R}^{d \times n}$, where $d$ is the tile size and $n$ is the number of tiles. For the reshape, we consider the dimensions in the order $f_{out}, f_{in}, h, w$ in our experiments. The goal is to factorize $\widetilde{\mathbf{W}}$ into the product of two matrices as follows:

$$\widetilde{\mathbf{W}} = \mathbf{CZ}, \tag{1}$$

where $\mathbf{C} \in \mathbb{R}^{d \times k}$ is the codebook, and $\mathbf{Z} \in \mathbb{R}^{k \times n}$ is the set of linear coefficients, or a latent variable ($k < d < n$). Following the standard PCA method, we factorize the zero mean version of $\widetilde{\mathbf{W}}$. With this decomposition, every column of $\widetilde{\mathbf{W}}$, denoted by $\widetilde{\mathbf{W}}_{:,i}$, is a linear combination of $k$ codebook vectors (columns of $\mathbf{C}$):

$$\widetilde{\mathbf{W}}_{:,i} = \sum_{j=1}^{k} Z_{ji} \mathbf{C}_{:,j}, \tag{2}$$

where $\mathbf{C}_{:,j}$ is the $j$-th column of $\mathbf{C}$. This decomposition problem is an instance of sparse PCA methods (Zou et al. (2006)). For network compression, we are additionally interested in quantized matrices $\mathbf{C}$ and $\mathbf{Z}$. The quantization operation for an arbitrary $\mathbf{C}$ and $\mathbf{Z}$ is defined as follows:

$$\mathbf{C}_q = Q_c(\mathbf{C}; \boldsymbol{s}_c, b_c), \tag{3}$$

$$\mathbf{Z}_q = Q_z(\mathbf{Z}; \boldsymbol{s}_z, b_z), \tag{4}$$

where $b_c, b_z$ are the quantization bit-widths, and $\boldsymbol{s}_c$ and $\boldsymbol{s}_z$ are the quantization scale vectors for $\mathbf{C}$ and $\mathbf{Z}$, respectively. We consider per-channel quantization, i.e., the quantization scale values are shared for each column of $\mathbf{C}$ and each row or column of $\mathbf{Z}$ (see section 3.2 for details on which is used when):

$$C_{q,ij} = \text{clamp}\left(\left\lfloor \frac{C_{ij}}{s_i} \right\rceil, 0, 2^{b_c} - 1\right) s_i, \tag{5}$$

$$Z_{q,ij} = \text{clamp}\left(\left\lfloor \frac{Z_{ij}}{s_j} \right\rceil, 0, 2^{b_z} - 1\right) s_j, \tag{6}$$

where $\lfloor \cdot \rceil$ is rounding to nearest integer, and clamp$(\cdot)$ is defined as:

$$\text{clamp}(x, a, b) = \begin{cases} a & x < a \\ x & a \le x \le b \\ b & x > b \end{cases}. \tag{7}$$

We refer to the problem of finding quantized factors $\mathbf{C}$ and $\mathbf{Z}$ with sparse $\mathbf{Z}$ as quantized sparse PCA problem. Once we obtain the factors $\mathbf{C}$ and $\mathbf{Z}$, we get the matrix $\widetilde{\boldsymbol{W}} = (\mathbf{CZ})$, which can be reshaped back to a convolutional layer. The reshaped convolutional layer for factors $\mathbf{C}, \mathbf{Z}$ is denoted by $[\mathbf{CZ}]$. It is well known in network compression literature that it is better to find the best factorization on the data manifold (Zhang et al. (2015); He et al. (2017); Stock et al. (2019)). Therefore, we solve the following optimization problem:

$$\mathbf{C}^*, \mathbf{Z}^* = \underset{\mathbf{C}_q, \mathbf{Z}_q}{\text{argmin}} \, \mathbb{E}_{(\boldsymbol{X}, \boldsymbol{Y}) \sim \mathcal{D}} \left( \left\| \boldsymbol{Y} - [\mathbf{C}_q \mathbf{Z}_q] * \boldsymbol{X} \right\|_F^2 \right) \tag{8}$$

$$\text{s.t.} \quad \mathbf{C}_q = Q_c(\mathbf{C}; \boldsymbol{s}_c, b_C)$$

$$\mathbf{Z}_q = Q_z(\mathbf{Z}; \boldsymbol{s}_z, b_Z)$$

$$\|\mathbf{Z}_q\|_0 \le S, \mathbf{Z} \in \mathbb{R}^{k \times n}, \mathbf{C} \in \mathbb{R}^{d \times k},$$

where the parameter $S$ controls the sparsity ratio of $\mathbf{Z}$, $\boldsymbol{X}$ and $\boldsymbol{Y}$ are the stored input output of the target layer in the original model, $\mathcal{D}$ is the data distribution, and $*$ is the convolution operation. $L_0$ norm is used for the constraint on the number of nonzero elements in the matrix $\mathbf{Z_q}$. We approximate the expected value of above optimization problem using a subset of the training data:

$$\mathbb{E}_{(\boldsymbol{X}, \boldsymbol{Y}) \sim \mathcal{D}} \left( \left\| \boldsymbol{Y} - [\mathbf{C}_q \mathbf{Z}_q] * \boldsymbol{X} \right\|_F^2 \right) \approx \frac{1}{m} \sum_{i=1}^m \left\| \boldsymbol{Y}_i - [\mathbf{C}_q \mathbf{Z}_q] * \boldsymbol{X}_i \right\|_F^2,$$

where $m$ is the number of samples used for optimization. Following the compression method introduced in Zhang et al. (2015), we use output of the previous compressed layer as $\boldsymbol{X}$ instead of the stored input to the layer in the original model. This approach aims at compensating for the error accumulation in deep networks.

### 3.1.1 Projected Gradient Descent Optimization Algorithm

The above optimization problem can be solved using an iterative projected gradient descent method. The projection step is a hard thresholding operation which project onto the space of sparse matrices. The optimization algorithm is given below.

1. **Initialization.** The codebook $\mathbf{C}$ is initialized with first $k$ left singular vectors of $\widetilde{\mathbf{W}}$. Given the SVD decomposition of $\widetilde{\mathbf{W}}$ as follows:

$$\widetilde{\mathbf{W}} = \mathbf{U\Sigma V}^\top, \tag{9}$$

---

**Algorithm 1** Projected Gradient Descent Optimization

---

**Require:** $\widetilde{\mathbf{W}}$
1: SVD of $\widetilde{\mathbf{W}} = \mathbf{U}\boldsymbol{\Sigma}\mathbf{V}^\top$,
2: $\mathbf{C} \leftarrow \mathbf{U}_k$
3: $\mathbf{Z} \leftarrow \mathbf{U}_k^\top \widetilde{\mathbf{W}}$
4: **while** Stopping criteria not met **do**
5:     **Gradient descent step: $\mathbf{C}, \mathbf{Z} \leftarrow$** gradient descent update on equation (10).
6:     $\mathbf{M} \leftarrow$ binary mask for largest $S$ values of $Q_z(\mathbf{Z}; \boldsymbol{s}_z, b_Z)$.
7:     **Hard thresholding step: $\mathbf{Z} \leftarrow \mathbf{Z} \odot M$.**
8: **end while**
9: **return:** $Q_c(\mathbf{C}; \boldsymbol{s}_c, b_C), Q_z(\mathbf{Z}; \boldsymbol{s}_z, b_Z)$.

---

we choose $\mathbf{C}^{(0)} = \mathbf{U}_k$, where $\mathbf{U}_k$ denotes the top $k$ left singular vectors of $\widetilde{\mathbf{W}}$. The latent matrix $\mathbf{Z}$ is initialized as the projection of $\widetilde{\mathbf{W}}$ onto the set of first $k$ singular vectors, $\mathbf{Z}^{(0)} = \mathbf{U}_k^\top \widetilde{\mathbf{W}}$.

2. **Gradient Descent.** At iteration $t$, the matrices $\mathbf{C}^{(t-1)}, \mathbf{Z}^{(t-1)}$ are updated by gradient descent on the following objective w.r.t. $\mathbf{C}$ and $\mathbf{Z}$:

$$\frac{1}{m} \sum_{i=1}^{m} \|\mathbf{Y}_i - [Q_c(\mathbf{C}; \boldsymbol{s}_c, b_C)Q_z(\mathbf{Z}; \boldsymbol{s}_z, b_Z)] * \mathbf{X}_i\|_F^2, \tag{10}$$

We use the straight through estimator (STE) (Bengio et al. (2013)) to compute the gradients of the quantization operation, i.e. we use the following gradient estimate for the rounding operation:

$$\frac{\partial \lceil p \rfloor}{\partial p} = 1.$$

The outcome of this step is denoted by $\mathbf{C}_{\text{gd}}^{(t)}, \mathbf{Z}_{\text{gd}}^{(t)}$.

3. **Hard thresholding.** After the gradient descent, we project $\mathbf{Z}_{\text{gd}}^{(t)}$ onto the space of sparse matrices. The projection is done using entrywise product of $\mathbf{Z}_{\text{gd}}^{(t)}$ with a binary mask matrix $\mathbf{M}^{(t)}$, namely $\mathbf{Z}_{\text{gd}}^{(t)} \odot M^{(t)}$. The mask matrix $M^{(t)}$ is obtained finding the support of $S$ largest entries of $Q_z(\mathbf{Z}_{\text{gd}}^{(t)}; \boldsymbol{s}_z, b_Z)$. This is the hard thresholding step. For the next iteration we set:

$$\mathbf{Z}^{(t)} = \mathbf{Z}_{\text{gd}}^{(t)} \odot M^{(t)}, \mathbf{C}^{(t)} = \mathbf{C}_{\text{gd}}^{(t)}.$$

4. Repeat the previous three steps using $\mathbf{Z}^{(t)}, \mathbf{C}^{(t)}$ until termination criteria is met (MSE error or a fixed iterations number). The final matrices are given be $Q_c(\mathbf{C}^{(t)}; \boldsymbol{s}_c, b_C), Q_z(\mathbf{Z}^{(t)}; \boldsymbol{s}_z, b_Z)$.

This iterative projection step has theoretical support for well behaved optimization problems. See Appendix A for more details.

### 3.1.2 RELATION TO OTHER APPROACHES

In this section, we discuss the connection with other existing methods.

(A) **Vector quantization.** If we replace the quantizers for $\mathbf{C}$ and $\mathbf{Z}$ by identity functions, i.e. $Q_c = \text{id}(\cdot)$, $Q_z = \text{id}(\cdot)$, then matrix $\mathbf{C}$ is in full precision, while $\mathbf{Z}$ encodes the indices such that

$$Z_{ij} = \delta_{im_i}, \tag{11}$$

where $m_i$ is the centroid index corresponding to tile $i$.

(B) **Additive vector quantization.** If $Q_c = \text{id}(\cdot)$, $b_z = 1$ (binary quantization), then

$$\widetilde{\mathbf{W}}_{:,i} = \sum_{j=1}^{k} \rho_{ji} \mathbf{C}_{:,j}, \tag{12}$$

where $\rho_{ji} \in \{0, 1\}$ are binary coefficients. Note that in the original work by Babenko & Lempitsky (2014), an additional constraint of $\sum_j \rho_{ji} = n_t$ is enforced, i.e. each $\mathbf{W}_{i,:}$ is a sum of a fixed number of terms.

(C) **Principle component analysis.** If $Q_c = \text{id}(\cdot)$, and $Q_z = \text{id}(\cdot)$, then

$$\widetilde{\mathbf{W}}_{:,i} = \sum_{j=1}^{k} Z_{ji} \mathbf{C}_{:,j}. \tag{13}$$

This case corresponds to SVD factorization of zero mean version of weight matrix $\widetilde{\mathbf{W}}$. For specific values of $n$ and $d$, the method is mathematically equivalent to the SVD decomposition methods for neural network compression (Denton et al. (2014); Jaderberg et al. (2014)). For example, if $n = f_{out}h$, and $d = f_{in}w$, then the method is equivalent to the spatial SVD decomposition by Jaderberg et al. (2014). In contrast to the above-mentioned approaches, we do not decompose the convolutional operation into two convolutional layers, which gives extra flexibility in the choice of PCA dimensionality $d$.

## 3.2 COMPRESSION RATIO

Without sparsity, the compression ratio is computed as follows. Let $L_o(\mathbf{W}) = f_{out} \times f_{in} \times h \times w \times 32$ denote the number of bits required to represent the original 32 bit floating point tensor $\mathbf{W}$, and $L_c(\mathbf{W}) = d \times k \times b_c + k \times n \times b_z$ the size of the quantized tensor factors $\mathbf{C}$ and $\mathbf{Z}$ of $\mathbf{W}$. The compression ratio $C_r$ is then defined as $C_r = \frac{L_o(\mathbf{W})}{L_c(\mathbf{W})}$.

Note that, since $k < d < n$, the latent matrix $\mathbf{Z}$ contributes more to the resulting model size than the codebook matrix $\mathbf{C}$. For example, let us assume $f_{out} = f_{in} = 256$, $h = w = 3$, $d = 256$, and $k = 128$. In this scenario, the original weight tensor $\mathbf{W}$ has $256 \times 256 \times 3 \times 3 = 589,824$ elements, the codebook matrix $\mathbf{C}$ has $256 \times 128 = 32,768$ elements and the latent matrix $\mathbf{Z}$ has $128 \times 2,034 = 260,356$ elements, almost 8 times as many as $\mathbf{C}$. For this reason, efforts to compress $\mathbf{Z}$, e.g., by using a lower bitwidth $b_c$ or pruning elements, will likely yield a better accuracy vs compression trade-off than focusing similar efforts on $\mathbf{C}$.

Additional compression is achieved by encoding the sparse matrix $\mathbf{Z}$ as follows. We assume the sparsity mask $\mathbf{M}$ is stored as a binary matrix. Then, we only store the nonzero values in $\mathbf{Z}$. At runtime, $\mathbf{Z}$ can be decoded by only reading values in $\mathbf{Z}$ for which the corresponding binary mask value is equal to 1. The total number of bits required to store $\mathbf{M}$ and the nonzero values in $\mathbf{Z}$ is $k \times n + (1 - r) \times k \times n \times b_z$, where $r = 1 - S/kn$ is the sparsity ratio of $\mathbf{Z}$ with $\|\mathbf{Z}\|_0 = S$. Note that this method only reduces memory footprint if $r > 1/b_z$.

The size of the compressed tensor must be adjusted for quantization parameters introduced by per-channel quantization. We assume that each quantization scale parameter is encoded using FP16. To account for this we add a quantization adjustment term $L_q(\mathbf{W})$ to $L_c(\mathbf{W})$, where $L_q = 2k \times 16$ is used for per-row quantization of $\mathbf{Z}$.

Finally, taking into account the adjustments for sparsity and quantization parameters, the size of a compressed tensor is computed as $L_c(\mathbf{W}) = d \times k \times b_c + k \times n + (1 - r) \times k \times n \times b_z + L_q(\mathbf{W})$.

## 4 EXPERIMENTS

In this section we describe two sets of experiments: one set to explore the effect of compression hyperparameters on compression-accuracy trade-offs, and one set of experiments to compare against various baseline methods.

### 4.1 EXPERIMENTAL SETUP

**Initial factorization** We start with a pre-trained model and perform the factorization as described in Section 3.1.1. We initialize the per-channel quantization scale using min-max quantization (Krishnamoorthi, 2018).

**Per layer data-aware optimization** To perform the data-aware optimization, we sample a single batch of 64 input and output activations for a layer. We split the data into a training set and a validation set (we keep one eighth of the data as the validation set). We keep track of the error on the validation set. During this step we use the Adam optimizer (Kingma & Ba (2014)) with learning rate $10^{-4}$ and weight decay $10^{-5}$.

**Sparsification** Sparsification is applied after per layer data-aware optimization in a form of hard thresholding. We consider one-shot and iterative thresholding. Specifically, for each layer, the quantized weights are sorted by absolute value, then a desired percentage of values is masked out. We do not prune weights that become zeros after the quantization step, instead, we add extra sparsity on top of accidental sparsity that occurs after the quantization. During the consequent fine-tuning stage, the sparsity mask is frozen.

**Fine-tuning** We follow Martinez et al. (2021) and fine-tune our models end-to-end using the target training set. Model-specific finetuning procedures are described below.

### 4.1.1 A NOTE ON COMPRESSION RATIO

In our experiments we leave the first convolutional weight tensor, all bias tensors, all batch normalization parameters, and the PCA centering vectors in FP32. We denote these uncompressible parameters as $L_u$. The sparsity ratio for a network is then computed as $C_r = \frac{\sum_{\mathbf{W}} L_o(\mathbf{W})}{L_u + \sum_{\mathbf{W}} L_c(\mathbf{W})}$

### 4.2 IMAGENET EXPERIMENTS

We evaluate our method on the ResNet18 and MobileNetV2 achitectures for ImageNet classification. We apply our method on pretrained versions of the model, where we take the pretrained weights from the PyTorch model zoo. In all our experiments we keep the tile size $d$ constant at 256, the bitwidth of the codebook matrix $\mathbf{C}$ constant at 4, and use per-channel scalar quantization of 4 bits for the fully connected classification layer weights. To achieve different compression ratios we vary the rank $k$ with values between 64-256, the bitwidth of the latent matrix $\mathbf{Z}$ between 3-6, and the extra sparsity of $\mathbf{Z}$ between 0% and 40%. Further experimental details can be found in Section B.1.

### 4.2.1 BASELINE METHODS

We compare our methods against scalar quantization by Esser et al. (2019), binary CNNs by Lin et al. (2017) (ABC-Net), the vector quantization method by Martinez et al. (2021) (PQF), the vector quantization method by Stock et al. (2019) (BGD), trained ternary quantization by Zhu et al. (2016) (TTQ). The results for scalar quantization are produced using LSQ Esser et al. (2019), with 4, 3 and 2-bit per-channel weight quantization and FP32 bit activations. For these experiments a weight decay of $10^{-4}$ and cosine learning rate decay to $10^{-2}$ of the original learning rate was used.

### 4.2.2 RESULTS

The Pareto dominating results of our method and comparison to the baselines can be found in Figure 2. The full set of (non Pareto dominating) results can be found in Figure 3a. The same results along with the values for rank, bitwidth and sparsity and resulting compression ratios can be found in Table 2 in the Appendix. In this figure we see that we outperform BGD, TTQ, ABC-Net and 3 and 2 bit scalar quantization by considerable margins. Furthermore, we outperform or match the strongest baseline at high compression ratios, PQF for compression ratios of 17x and higher, and are on par with 4-bit scalar quantization for compression ratios under 17x. Lastly, we are the only method to achieve SOTA compression-accuracy trade-offs over the full range from 5x to 40x compression.

### 4.3 ABLATION STUDIES

### 4.3.1 HARD THRESHOLDING METHODS

In this section, we study the impact of the hard thresholding method and the stopping criterion for data-aware optimization. Algorithm 1 applies the hard-thresholding step at each iteration. However, we can also apply it only after the end of gradient descent iterations. We call this one-shot hard-thresholding. The method has precedence in non-convex sparse optimization (Shen (2020)). We consider two different stopping criteria. The first is using the fixed number of gradient descent iterations, e.g., 30, 100, or 1000. The second is based on achieving the MSE error threshold on the validation set. We split the data used for per-layer optimization into a training set and a validation set. We stop the iterations as soon as the error on the validation set is not decreasing for more

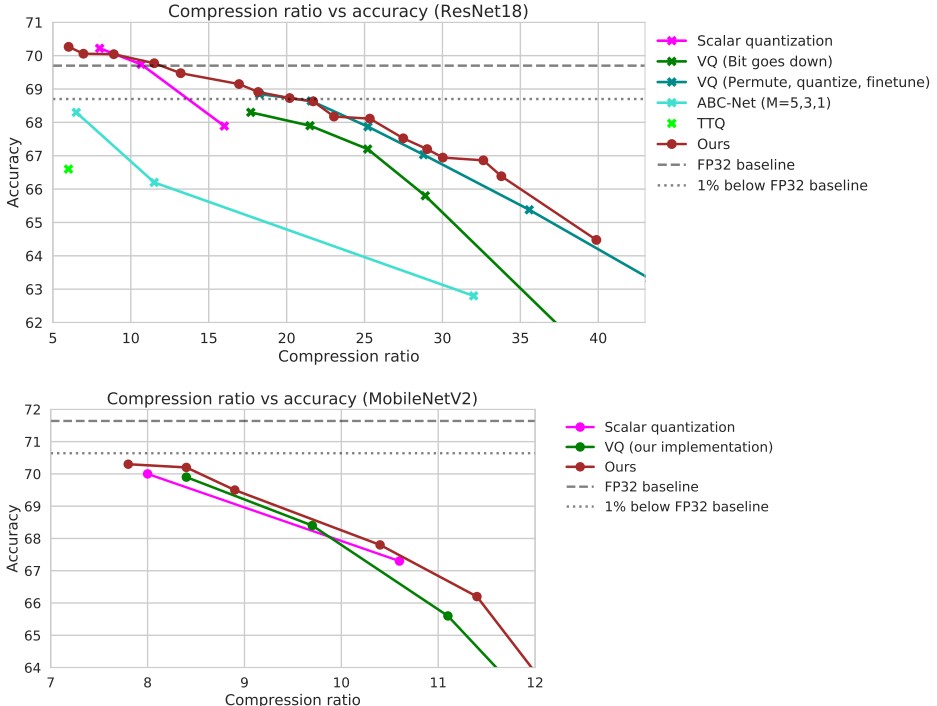

Figure 2: Results for Resnet18 and MobileNetV2 trained on ImageNet

than two iterations. Finally, we compare both methods with one-shot hard-thresholding without data-aware optimization.

We present the results for Resnet18 in the Table 1. Three different levels of sparsity are considered for a model with 4 bit latent. We report the validation accuracy of the model before and after fine-tuning. The results suggest that the most important aspect of data-aware optimization is the stopping criterion. The best pre-finetuning results are obtained by using iterative hard thresholding with the least number of iterations, namely 30. The second best method is one-shot iterative hard thresholding with the stopping criterion based on the validation set. These results suggest that it is important to limit the number of iterations of per layer optimization in order to prevent the method from over-fitting to the layer input and output data sample. This intuition is further supported by observing the lowest pre-finetuning accuracy for hard thresholding with the highest number of iterations, namely 1000.

However, overall benefits of per layer optimization methods become marginal after the compressed model is fine-tuned. In various experiments, we observed an 0.1-0.3% improvement in the validation accuracy compared to using one-shot hard thresholding without any data-aware optimization. Therefore, in our further full model experiments, we used one-shot hard thresholding with the stopping criterion based on the validation set as a method of low computational complexity, which gives nearly the best results compared to the other methods.

### 4.3.2 QUANTIZATION VS PRUNING VS RANK

In our method, compression can be achieved by lowering rank, lowering the bit-width for latent matrices (hence referred to as 'bit-width'), or increasing extra sparsity. Thus, similar compression ratios can be achieved with different values for rank, bit-width and sparsity. For example, for a given model, we can retain the same compression ratio by increasing rank and decreasing bit-width to compensate. In this section we investigate whether we can discover patterns that would help guide compression hyperparameter search.

In Figure 3a we show a scatter plot with the full set of results. In this figure, we see that for low compression ratios (below 14x) only minor improvements can be achieved by tweaking compression hyperparameters. For example, around 10x compression even the worst performing compression hy-

| Hard threshold-ing method | Stopping. criterion | Pre-FT accuracy | | | Post-FT accuracy | | |
|---|---|---|---|---|---|---|---|
| | | 10% sparsity | 20% sparsity | 30% sparsity | 10% sparsity | 20% sparsity | 30% sparsity |
| One-shot HT | val. set | 63.2 | 58.9 | 47.3 | **69.1** | 68.7 | **68.1** |
| One-shot HT | 100 iter. | 59.4 | 55.5 | 45.2 | 68.8 | 68.4 | 68.0 |
| Iterative HT | 30 iter. | **64.5** | **63.0** | **58.8** | **69.1** | **68.8** | 68.0 |
| Iterative HT | 100 iter. | 62.0 | 60.7 | 57.3 | 69.0 | 68.7 | **68.1** |
| Iterative HT | 1000 iter. | 55.0 | 55.4 | 54.1 | 68.5 | 68.3 | 67.9 |
| No data-aware opt. | - | 58.5 | 53.3 | 37.8 | 69.0 | 68.7 | 67.9 |

Table 1: Resnet18 ablation on the impact of the hard thresholding method and the stopping criterion on top-1 validation accuracy of the compressed model. Iterative hard-thresholding with 30 iterations shows the best results among all the other methods, however, the benefit vanishes after fine-tuning.

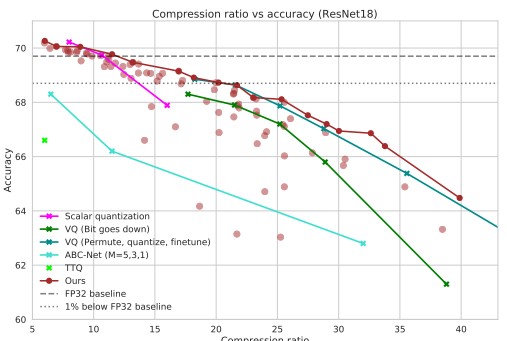

(a) All results for ResNet18, including non-Pareto dominating results

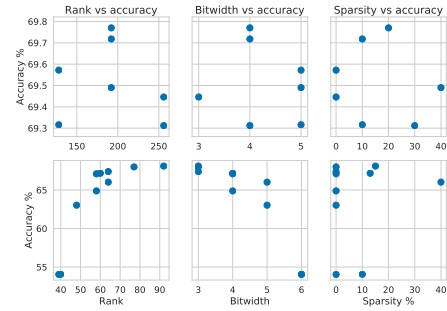

(b) Effect of rank, bit-width and sparsity for models near 11x (top row) and 25x (bottom row) compression ratios.

Figure 3: Full results and effect of rank, bit-width and sparsity.

perparameters are still on par with quantization to 3 bits (10.67x compression). However, for higher compression ratios the compression hyperparameters have a larger impact on model performance.

In Figure 3b we show the effect of rank, bit-width and sparsity ratios on a set of models with a low (near 11x, top row) and a high (near 25x, bottom row) compression ratio. Again we see little effect (roughtly 0.5%) for the lower compression ratio. However, for the more aggressively compressed models in the bottom row, we see that increasing rank while lowering bit-width to compensate has a strong positive effect on model performance. Finally, in both plots we see no clearly discernible trend in the effect of compression ratio on model accuracy.

Based on these results we conclude that, at high compression ratios, increasing rank and decreasing bit-widths can improve results. Sparsity can be used to further fine-tune the target compression ratio.

## 5 CONCLUSION

We presented a weight compression method based on quantized sparse PCA which unifies SVD-based weight compression, sparse PCA, and vector quantization. We tested our method on ImageNet classification and show compression-accuracy trade-offs which are state-of-the-art or on par with strong baselines, and demonstrated that our method is the only method to achieve SOTA compression-accuracy trade-offs in both low and high compression regimes. Lastly, we investigated whether low rank approximations, quantization, or sparsity yield the best efficiency-accuracy trade-offs, and found that at high compression ratios, increasing rank and decreasing bit-width can improve results. For future work, we plan to investigate methods for automated rank, bit-width, and sparsity ratio selection in order to further improve the accuracy and reduce the hyper-parameter search time.

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

## A   HARD THRESHOLDING FOR SPARSE OPTIMIZATION

The sparse optimization method sketched in this paper is based on one-shot or alternating applications of a projection step onto the space of sparse vectors. In this section, we provide some theoretical supports for this approach. Consider the following optimization problem:

$$\min_{\boldsymbol{x}\in\mathcal{D}} f(\boldsymbol{x}). \tag{14}$$

The set $\mathcal{D}$ is a generic set. The projection function on $\mathcal{D}$ is defined as

$$\Pi_{\mathcal{D}}(\boldsymbol{z}) = \arg\min_{\boldsymbol{x}\in\mathcal{D}} \|\boldsymbol{x} - \boldsymbol{z}\|_2 .$$

When the set $\mathcal{D}$ is the set of sparse vectors denoted by $\Sigma_s(\mathbb{R}^n)$ with sparsity order $s$, the projection function is hard thresholding. For a vector $\boldsymbol{x}$, the hard thresholding function $H_s(\boldsymbol{x})$ yields a vector with at most $s$ non-zero entries, which are the $s$ entries of $\boldsymbol{x}$ with largest absolute value.

The following proposition provides a general condition under which the optimization problem (14) can be solved using iterative projection methods.

**Proposition A.1.** Consider the optimization problem equation 14 and assume that $\boldsymbol{x}_\star$ is the minimizer of the problem. Consider an iterative optimization algorithm with the initial point $\boldsymbol{x}_0$ and the iteration $t$ defined by:
$$x_{t+1} = \Pi_\mathcal{D}(\boldsymbol{x}_t - \Delta_f(\boldsymbol{x}_t)),$$
where $\Delta_f(.)$ is a function such that $\boldsymbol{x} - \Delta_f(\boldsymbol{x})$ is $L$-Lipschitz, and $\Delta_f(\boldsymbol{x}_\star) = 0$. Then we have:
$$\|\boldsymbol{x}_t - \boldsymbol{x}_\star\|_2 \le (2L)^t \|\boldsymbol{x}_0 - \boldsymbol{x}_\star\|_2.$$
In particular, if $L \le 1/2$, the algorithm converges to the minimizer $\boldsymbol{x}_\star$ as $t \to \infty$.

*Proof.* First, for an arbitrary set $\mathcal{D} \subseteq \mathbb{R}^n$, the projection operation onto the set $\Pi_\mathcal{D}(\cdot)$ and any $\boldsymbol{x} \in \mathcal{D}$ and $\boldsymbol{z} \in \mathbb{R}^n$, we have the following inequality
$$\|\Pi_\mathcal{D}(\boldsymbol{z}) - \boldsymbol{x}\|_2 \le 2\|\boldsymbol{z} - \boldsymbol{x}\|_2$$
We can obtain this inequality in two steps. We first use triangle inequality:
$$\|\Pi_\mathcal{D}(\boldsymbol{z}) - \boldsymbol{x}\|_2 \le \|\boldsymbol{z} - \boldsymbol{x}\|_2 + \|\Pi_\mathcal{D}(\boldsymbol{z}) - \boldsymbol{z}\|_2.$$
For the last term, we use the definition of projection operation that for each $\boldsymbol{x} \in \mathcal{D}$, the projection of $\boldsymbol{z}$ to $\mathcal{D}$ minimizes its distance to the points in $\mathcal{D}$:
$$\|\Pi_\mathcal{D}(\boldsymbol{z}) - \boldsymbol{z}\|_2 \le \|\boldsymbol{z} - \boldsymbol{x}\|_2.$$
Consider the iteration step $t + 1$. We have:
$$\|\Pi_\mathcal{D}(\boldsymbol{x}_t - \Delta_f(\boldsymbol{x}_t)) - \boldsymbol{x}_\star\|_2 \le 2\|\boldsymbol{x}_t - \Delta_f(\boldsymbol{x}_t) - \boldsymbol{x}_\star\|_2 \le 2L\|\boldsymbol{x}_t - \boldsymbol{x}_\star\|_2,$$
which implies
$$\|\boldsymbol{x}_t - \Delta_f(\boldsymbol{x}_t) - \boldsymbol{x}_\star\|_2 \le (2L)^{t+1}\|\boldsymbol{x}_0 - \boldsymbol{x}_\star\|_2.$$
$\square$

Some remarks are in order. First, the assumption $\Delta_f(\boldsymbol{x}_\star) = 0$ means that $\boldsymbol{x}_\star$ is a stationary point of the algorithm. This holds particularly for gradient descent based methods where $\Delta_f(\boldsymbol{x}_\star) = \alpha \nabla f(\boldsymbol{x})$ and $\boldsymbol{x}_\star$ is a stationary point of $f$.

The Lipschitz property of $\boldsymbol{x} - \Delta_f(\boldsymbol{x})$ holds in many situations. As an example, consider the function $f(\cdot)$ as a simple mean squared error (MSE) loss, i.e., $f(\boldsymbol{x}) = \|\boldsymbol{y} - \boldsymbol{A}\boldsymbol{x}\|_2^2$. In this case, we have:
$$\boldsymbol{x} - \Delta_f(\boldsymbol{x}) = \boldsymbol{x} - \alpha\boldsymbol{A}\boldsymbol{A}^\top\boldsymbol{x} = (\boldsymbol{I} - \alpha\boldsymbol{A}\boldsymbol{A}^\top)\boldsymbol{x},$$
which satisfies $L$-Lipschitz property with $L$ bounded by the spectral norm of $\boldsymbol{I} - \alpha\boldsymbol{A}\boldsymbol{A}^\top$. In many situations, when this spectral norm is restricted to the set $\mathcal{D}$, it can be controlled to be less than $1/2$. A notable example, closely related to our problem, is the sparse recovery problem where one can prove recovery guarantee for iterative hard thresholding methods when the matrix $\boldsymbol{A}$ satisfies *restricted isometry property*. See Foucart & Rauhut (2013) for a comprehensive exposition of the topic including iterative hard thresholding for linear problems. Hard thresholding methods are widely used in sparse optimization literature. Particularly relevant to our problem is the one-bit sparse recovery problem, where iterative hard thresholding is used Jacques et al. (2013).

# B  RESNET18 EXPERIMENTS

## B.1  EXPERIMENTAL DETAILS

We run a small number of pilot experiments for 5 epochs, to find good settings for the optimizer, learning rate, and learning rate schedule. We then run a large number of 5 epoch experiments to find good compression-accuracy trade-offs. For the best performing compression-accuracy trade-offs we then run longer experiments of 25 epochs, with the two best optimization schedules. We use a batch size of 64 for Resnet18, use cosine learning rate decay to $10^{-3}$ of the initial learning rate, and turn off weight decay.

## B.2  PARETO DOMINATING RESULTS

## B.3  ABLATION STUDY RESULTS

| K | D | bw | mpfd | cr_sparse | eval_score |
|---|---|---|---|---|---|
| 256 | 256 | 5 | 0.00 | 6.01 | 70.26 |
| 256 | 256 | 5 | 0.20 | 6.96 | 70.06 |
| 256 | 256 | 4 | 0.20 | 8.91 | 70.04 |
| 192 | 256 | 4 | 0.20 | 11.50 | 69.77 |
| 192 | 256 | 3 | 0.00 | 13.20 | 69.47 |
| 128 | 256 | 3 | 0.00 | 16.95 | 69.15 |
| 128 | 256 | 3 | 0.00 | 18.17 | 68.91 |
| 128 | 256 | 4 | 0.40 | 20.19 | 68.73 |
| 128 | 256 | 3 | 0.20 | 21.70 | 68.63 |
| 92 | 256 | 3 | 0.15 | 23.04 | 68.17 |
| 92 | 256 | 3 | 0.15 | 25.34 | 68.11 |
| 100 | 256 | 3 | 0.12 | 27.49 | 67.52 |
| 64 | 256 | 3 | 0.00 | 29.02 | 67.20 |
| 92 | 256 | 3 | 0.15 | 30.02 | 66.94 |
| 64 | 256 | 3 | 0.20 | 32.62 | 66.86 |
| 64 | 256 | 3 | 0.00 | 33.78 | 66.37 |
| 64 | 256 | 3 | 0.20 | 39.88 | 64.48 |

Table 2: All Pareto dominating results for ResNet18

| | compression ratios near 11x | | | | | | compression ratios near 25x | | | | |
|---|---|---|---|---|---|---|---|---|---|---|---|
| K | D | bw | sparsity | comp. ratio | accuracy | K | D | bw | sparsity | comp. ratio | accuracy |
| 256 | 256 | 4 | 0.3 | 10.87 | 69.31 | 39 | 256 | 6 | 0.00 | 25.97 | 54.04 |
| 128 | 256 | 5 | 0.1 | 11.41 | 69.32 | 40 | 256 | 6 | 0.10 | 25.52 | 54.06 |
| 256 | 256 | 3 | 0.0 | **11.77** | 69.45 | 48 | 256 | 5 | 0.00 | 25.26 | 63.03 |
| 192 | 256 | 5 | 0.4 | 11.07 | 69.49 | 58 | 256 | 4 | 0.00 | 25.57 | 64.89 |
| 128 | 256 | 5 | 0.0 | 11.24 | 69.57 | 64 | 256 | 5 | 0.40 | 25.57 | 66.03 |
| 192 | 256 | 4 | 0.2 | 11.19 | **69.77** | 58 | 256 | 4 | 0.00 | 25.57 | 67.11 |
| 192 | 256 | 4 | 0.1 | 10.56 | 69.72 | 60 | 256 | 4 | 0.13 | 25.45 | 67.18 |
| | | | | | | 64 | 256 | 3 | 0.00 | **26.07** | 67.40 |
| | | | | | | 77 | 256 | 3 | 0.00 | 25.56 | 68.00 |
| | | | | | | 92 | 256 | 3 | 0.15 | 25.34 | **68.11** |

Table 3: Various hyperparameter settings and resulting compression ratios and accuracies for compression ratios near 11x and 25.5x

