# OpenReview forum: "Quantized sparse PCA for neural network weight compression"
_ICLR.cc/2022/Conference — ICLR 2022 Submitted_

### Official Review · Reviewer_iyVU · 2021-10-21

**Correctness:** 4
**Technical Novelty And Significance:** 3
**Empirical Novelty And Significance:** 3
**Recommendation:** 8
**Confidence:** 3

**Main Review:**

# Strenghts
S1. The paper is clearly written and the proposed method and results are convincing. The method is simple yet seems to work well.

S2. The topic of weight compression should be of interest to a large proportion of the deep learning/ML community.

S3. Although I’m not familiar with the literature in this area, the authors seem to do a good job of discussing related work (in Sec. 2) and later on also discussing how their proposal relates to other previous methods (in Sec. 3.1.2).

# Weaknesses
W1. I mentioned the simplicity of the method as a strength. However, the simplicity of the method comes from the fact that it just combines several previous ideas (PCA, sparsity, quantization) into a new method, which could be seen as incremental/less exciting.

# Questions
Below are some questions I have for the authors.

Q1. The paper discusses the accuracy/compression trade off, but focuses less on time/computational cost. What can you say about the time it takes to compress a network? Is it negligible compared to, e.g., the time it takes to train the network?

Q2. Does having the weights in the sparse quantized PCA format increase the inference time compared to the non-compressed network?

Q3. How is the reshaping of the weight tensor $W$ into $\tilde{W}$ done? Do you always put $f_{\text{out}}$ along the rows, and $f_{\text{in}}$, $h$, $w$ along the rows? You should mention this in the paper.

Q4. What does the $\lfloor \cdot \rceil$ notation in Eqs. (5) and (6) signify? Round to nearest integer?

Q5. Should it be $2^{b_c}$ in Eq. (5) and $2^{b_z}$ in Eq. (6) rather than just $2^b$?

Q6. At end of 1st paragraph in Sec. 4.3.1, you mention doing “one-shot hard-thresholding without data-aware optimization.” What do you mean by “without data-aware optimization”? Are you just doing an SVD of $\tilde{W}$ and then quantizing the $C$ and $Z$ matrices?

Q7. How can quantization be implemented in practice? Do you implement your experiments in Python?

Q8. What does “MAC operations” stand for (mentioned in the introduction)?

# Typos, other minor things
T1. Above Eq. (11), “while $Z$ encodes the of indices” should be “while $Z$ encodes the indices.”

T2. Sec. 3.2, 2nd paragraph: Should the size of $W$ be $256 \times 256 \times 3 \times 3$ rather than $256 \times 256 \times 9 \times 9$ since $w=h=3$?

T3. Sec. 4.3.2, 2nd paragraph, 1st sentence, there’s a period missing before “In this figure”.

T4. In 3rd sentence after the proof of Proposition A.1, should it be $\Delta_{f}(x) = \alpha \nabla f(x)$, i.e., without a $\star$ on the $x$ on the left hand side?

T5. In Sections B.2 and B.3 in the appendix, it would be nice if you at least added one sentence per section that refers to the relevant table so that we can know which table belongs in which section. Right now, the section headings aren’t really doing much.


**Summary Of The Paper:**

The paper proposes a method for compression of neural network weights. The proposal turns weight tensors into matrices, factorizes these matrices into a rank-k factorization via PCA, applies quantization to the factor matrices, and additionally makes the right (latent) matrix sparse. An algorithm is presented, with two different thresholding options (per-iteration, or one-shot). In experiments, these ideas lead to an accuracy/compression tradeoff which is either competitive with or better than previous state-of-the-art across all compression ratios investigated.

**Summary Of The Review:**

The paper is well written and covers a topic that should be of broad interest to the deep learning/ML community. The authors do a good job of putting their proposal in context and explaining previous work. The experiment results are encouraging. Overall, a nice paper that I think is suitable for publication in ICLR.

---

> ### Author Response · Authors · 2021-11-13
> **Answer**
>
> We thank the reviewer for the detailed feedback. We answer the comments and suggestions below.
>
> **W1**. As opposed to a straightforward combination, e.g. applying quantization on existing SVD decomposition, we solve the joint optimization problem (8). This problem of jointly optimizing the quantized values and the sparsity mask is a non-convex integer program and is NP-hard. Practically, it is challenging to solve. To the best of our knowledge, this type of problem was never considered in the literature. We look at the problem on the data manifold and solve it by introducing the alternating optimization method (Algorithm 1). The challenge of dealing with discrete quantized values is tackled using a straight-through estimator (STE).
>
> In order to compare the solution obtained with our method to a direct combination of the two methods (simply quantizing SVD-compressed network), we present the results in Table 1. For example for 30\% sparsity (see the pre-FT results) we obtain a network with 63.0\% top-1 accuracy (see iterative HT with 30 iterations) while the naive combination of the methods gives a network with 53.3\% accuracy (see "no data-aware opt." row).
>
> **Q1.** As our method is based on per layer weight optimization (Algorithm 1), its computational cost is much lower than network training/fine-tuning. Practically, it takes 1-2 minutes to compress an ImageNet model on a GPU.
>
> **Q2.** The sparse representation is only used when the sparsity level is higher than $1/b$ to ensure the total size of weights in memory does not increase. In practice, this holds for most of the layers in our experiments.
>
> **Q3.** In our experiments, we flatten the weight tensor in the order of dimensions $f_{out}, f_{in}, h, w$. Then we reshape it into $n \times d$, where $d$ is PCA dimensionality and $n$ is the number of tiles. We clarified it in the paper.
>
> **Q4.** Yes, this notation signifies rounding to the nearest integer. We added a clarification on this to the paper.
>
> **Q5.** Correct. Fixed in the paper, thank you for pointing it out.
>
> **Q6.** Exactly. No optimization is performed in this case.
>
> **Q7.** The experiments are implemented in PyTorch. We define a custom gradient operation for the quantizers where we use a straight-through estimator.
>
> **Q8.** By MAC we mean multiply-accumulate operation. We added a clarification to the paper.
>
> Thank you for mentioning the typos, we corrected the paper accordingly.

---

> > ### Comment · Reviewer_iyVU · 2021-11-16
> > **Thank you for the response**
> >
> > Thank you for your response.
> >
> > To clarify my question Q2: If the NN is e.g. used for classification, does it take longer to classify a single input using a compressed network vs an uncompressed network? Basically, does a forward pass in the compressed NN take longer than a forward pass in the uncompressed NN? It seems like it should take more time, since the weight tensors have to be reconstructed from the factors and then reshaped into the correct shape. This additional cost may be negligible in practice, but I was just curious to hear if you looked at this.
> >
> > After reading the other reviews and author responses, I'm leaving my score as it is for now.

---

> > > ### Author Response · Authors · 2021-11-19
> > > **Answer**
> > >
> > > While the main intention of our method is to reduce the model size and memory bandwidth, we confirm that the compute cost increases with our approach.
> > >
> > > The complexity of a forward pass for a convolutional layer is $f_{in}f_{out}hwp$, where $f_{in}$ and $f_{out}$ are the number of input and output channels, $h \times w$ is the spatial size of the filter, and $p$ is the number of pixels in the input feature map assuming "same" padding. The overhead for the filter reconstruction is the product of two matrices using $f_{in} f_{out}hwk$ multiply-add operations (MACs), where $k$ is PCA rank. So the relative overhead in the number of operations is $k/p$. Practically, for Resnet18, a value of $k$ equals for example 128, while the spatial size of the feature map for an ImageNet goes down from $56\times 56$ to $7\times7$ in the later layers. So, the relative overhead for a single forward pass is varying between 4\% and 261\% extra operations for different layers. While the total relative overhead for the full model is 85\% extra MACs. However, with an increasing batch size or when doing multiple inferences this because neglectable.
> > > Please also note that the above comparison is based on MACs, however in our approach, most computations contain low bit tensors. If we do the same comparison in bit operations (BOPs) the overhead is significantly reduced. We are going to add a clarification on the computational overhead to the paper.

---

> > > > ### Comment · Reviewer_iyVU · 2021-11-19
> > > > **Thank you for the clarification.**
> > > >
> > > > Thank you for the clarification. I think adding this to the paper would be nice.

---

### Official Review · Reviewer_zn4a · 2021-10-29

**Correctness:** 3
**Technical Novelty And Significance:** 2
**Empirical Novelty And Significance:** 2
**Recommendation:** 5
**Confidence:** 3

**Main Review:**

Strengths:
The authors empirically demonstrate that their method is the only method to achieve SOTA compression-accuracy trade-offs in both low and high compression regimes.

Weakness:
1. The method seems to be a simple combination of tensor factorization and vector quantization methods. The idea of using (sparse) PCA in vector quantization methods has also been studied, e.g., in Babenko & Lempitsky (2014). Therefore, the novelty in this submission seems to be quite limited.
2. The authors should provide more explanations about the motivation and benefits of assuming the factor matrix $\mathbf{Z}$ to be sparse. In addition, the authors should clarify that Z is sparse in what sense? For example, row-sparse, column-sparse, or sparse while considering $\mathbf{Z}$ as a $(kn)$-dimensional vector? In (8), the norm $\||\cdot\||_0$ is used without any explanation.
3. The authors should be careful to claim that their method for factorizing $\tilde{\mathbf{W}}$ in (1) is sparse PCA (ignoring the quantization). For conventional sparse PCA, the factor matrix $\mathbf{C}$, instead of $\mathbf{Z}$, is assumed to be column-sparse.
4. The gradient descent update on (10) is important and should be clearly presented in this submission, rather than simply citing another paper.
5. All source codes required for conducting experiments should be included in a code appendix.

Other comments:
1. The terminology "quantized sparse PCA" should be fixed, and the authors should not also use "sparse quantized PCA" arbitrarily (e.g., in the second dot point of the contributions).
2. The relation $k < d < n$ should be made clear when these parameters first appear in Section 3.1 (instead of mentioning it in Section 3.2).
3. In Algorithm 1, line 5, equation 10 should be equation (10). In Appendix A, problem 14 should be problem (14).

**Summary Of The Paper:**

This paper introduces a novel method of weight compression. Weight tensors are stored as sparse, quantized matrix factors, and the underlying matrix factorization problem can be considered as a quantized sparse PCA problem and be solved through iterative projected gradient descent methods. The authors' method  is applicable to both moderate and extreme compression regimes, and is claimed to achieve or be on par with state-of-the-art trade-offs between accuracy and model size.

**Summary Of The Review:**

Although the method proposed by the authors seems to be empirically promising, it seems to be a direct combination of existing methods, and the novelty seems to be limited. The authors provide no theoretical guarantee/intuition for their method. In addition, some terminologies/notations are vague or confusing. My impression is that this submission requires some major revisions.

---

> ### Author Response · Authors · 2021-11-13
> **Answer**
>
> We thank the reviewer for their comments and suggestions. We give our replies below.
>
> **W1.** Our method can be viewed as a generalization over SVD compression, sparse PCA, and vector quantization. Depending on the choice of quantizers and sparsity constraint we can arrive at any of three methods while we depart from a single unified formulation as we show in section 3.1.2. While additive quantization Babenko \& Lempitsky (2014) only consider sums of a fixed number of codewords for vector quantization which is a very restricted version of our method. Our formulation yields more expressive linear combinations which are in agreement with our experiments.
>
> As opposed to a straightforward combination, e.g. applying quantization on existing SVD decomposition, we solve the joint optimization problem (8). This problem of jointly optimizing the quantized values and the sparsity mask is a non-convex integer program and is NP-hard. Practically, it is challenging to solve. To the best of our knowledge, this type of problem was never considered in the literature. We look at the problem on the data manifold and solve it by introducing the alternating optimization method (Algorithm 1). The challenge of dealing with discrete quantized values is tackled using a straight-through estimator (STE).
>
> In Table 1 we present the results on our joint optimization method versus the naive baseline  (simply quantizing SVD-compressed network). For example for 30\% sparsity (see the pre-FT results) we obtain a network with 63.0\% top-1 accuracy using the joint optimization (see iterative HT with 30 iterations) while the naive combination of the methods gives a network with 53.3\% accuracy (see "no data-aware opt." row).
>
> **W2.** Our motivation for sparsifying matrix Z is inline
> with previous works on sparse PCA. We represent every row of weight matrix Z using a sparse linear combination of codewords. Practically, factor Z has a larger size in memory, so it is more important to compress Z versus compressing the codebook C (we explain the details in section 3.2).
> Regarding sparsity, we penalize the number of nonzero elements in Z as a (kn)-dimensional vector. We added a clarifying note to our formula which introduces the L0 norm.
>
> **W3.** In our work, we follow the conventional sparse PCA in the sense that we use dense codebook $C$ and a sparse set of linear coefficients $Z$. Although the order of $C$ and $Z$ is indeed reversed, i.e. the first factor would be sparse in the conventional PCA while the second factor would be dense. Strictly speaking, we perform sparse PCA on $W^T$, and should use $W^T$ in our notation, however, we deliberately omit the transposition to avoid unnecessary notation clutter (we assume the matrix W can always be transposed prior to the computation if needed).
>
> **W4.** The STE gradient for the rounding operation is defined as $\frac{\partial \lceil p\rfloor}{\partial p}=1$, we clarified this in the paper.
>
> **W5.** We agree that publishing the source code would help to ensure the reproducibility of our method. And we plan to release the source code upon acceptance. As a side note, this should not be a limiting factor for the paper according to the conference guidelines.
>
> **Reply to summary.** We can not provide theoretical guarantees for this non-convex problem, however, we give intuition on the convergence of Algorithm 1 in a simplified setting in Appendix A.
>
> **Other comments.** Thank you for the suggestions provided in "other comments", we addressed each of them in the updated manuscript accordingly.

---

> > ### Comment · Reviewer_zn4a · 2021-11-24
> > **Thanks for the response**
> >
> > For the response to W4, I don't think "The STE gradient for the rounding operation is defined as..." is a clarification. I expect the authors to write down the update rule for $\mathbf{C}$ and $\mathbf{Z}$ explicitly and clearly.
> >
> > For the response to W5, I don't understand why the authors cannot upload the source code during this reviewing phase. I apologize for my late response, but the authors can still reply to me with sharing a link to an anonymous repository for the code.

---

> > > ### Author Response · Authors · 2021-11-26
> > > **Answer**
> > >
> > > **W4.** First of all, we admit there is a mistake in the range limits in quantizers (5) and (6). As elements of $\mathbf{C}$ and $\mathbf{Z}$ are not required to be positive, the correct definition is (also different quantization scales are used for $\mathbf{C}$ and $\mathbf{Z}$):
> > >
> > > $C_{q,ij}=\mbox{clamp} \left( \left\lfloor \frac{C_{ij}}{s_{z,i}} \right\rceil , -2^{b_c-1}, 2^{b_c-1} - 1\right)s_{c,i},$
> > >
> > > $Z_{q,ij}=\mbox{clamp} \left( \left\lfloor \frac{Z_{ij}}{s_{z,j}} \right\rceil , -2^{b_z-1}, 2^{b_z-1} - 1\right)s_{z,j}$,
> > >
> > > We use the sequence of the following variables in order to calculate the derivative w.r.t $\mathbf{C}$ and $\mathbf{Z}$:
> > >
> > > $F= \frac{1}{m}\sum_{i=1}^m ||Y_i- O_i]||^2_F$,
> > >
> > > $F_i=||Y_i- O_i||^2_F$,
> > >
> > > $O_i=\tilde{W}*X_i$,
> > >
> > > $\tilde{W}=C_{q} Z_{q}$.
> > >
> > > The derivatives of interest are defined as follows:
> > >
> > > $\frac{\partial F}{\partial O_i}=2(Y_i-O_i)$,
> > >
> > > $\frac{\partial \tilde{W}}{\partial C_q}= Z_q^T$,
> > >
> > > $\frac{\partial \tilde{W}}{\partial Z_q}= C_q^T$.
> > >
> > > The derivatives of $\mathbf{C}_q$ can be expressed as follows (as Latex cases are not supported in this discussion, we have to use single line expressions):
> > >
> > > $\frac{\partial C_{q,ij}}{\partial C_{ij}}=1\mbox{ if } -s_{c,i}2^{b_c-1} \leq C_{ij} \leq s_{c,i}(2^{b_c-1}-1) \mbox{, 0 otherwise}$;
> > >
> > > $\frac{\partial C_{q,ij}}{\partial s_{c,i}}=-\frac{C_{ij}}{s_{c,i}}+\lfloor\frac{C_{ij}}{s_{c,i}}\rceil\mbox{ if  }-2^{b_c-1}\leq\frac{C_{ij}}{s_{c,i}}\leq 2^{b_c-1}-1, -2^{b_c-1}\mbox{ if } \frac{C_{ij}}{s_{c,i}} < -2^{b_c-1}, 2^{b_c-1}-1\mbox{ if } \frac{C_{ij}}{s_{c,i}} > 2^{b_c-1}-1$.
> > >
> > > Note that in the equations above, we used STE estimator for the rounding operation, i.e. the gradient of the quantizer is computed as if the rounding operation is bypassed in the backward pass.
> > > Using the equations above, the derivative of the objective $F$ w.r.t. $\mathbf{C}$ can be expressed as follows:
> > >
> > > $\frac{\partial F}{\partial C}=\sum_{i=1}^{m} \frac{\partial F_i}{\partial O_i}\frac{\partial O_i}{\partial \tilde{W}}\frac{\partial \tilde{W}}{\partial C_q}\frac{\partial C_q}{\partial C}$.
> > >
> > > The gradients of $Z$ can be expressed in a similar manner. We are ready to include the full derivation and the corrections to the paper once it is possible to upload a new revision.
> > >
> > > **W5.** We understand that having the source code available can aid the review process. Unfortunately, open-sourcing anonymous source code is not equally easy for every research institute due to legal and intellectual property considerations. In our case, the institute’s process did not enable us to release the source code on time for review, but we plan to release to code with the final paper.

---

### Official Review · Reviewer_rRp9 · 2021-11-01

**Correctness:** 2
**Technical Novelty And Significance:** 2
**Empirical Novelty And Significance:** 1
**Recommendation:** 1
**Confidence:** 5

**Main Review:**

1. I request the authors to discuss a significant low-rank factorization technique used for deep neural network gradient compression---PowerSGD by Vogels et al. NeuRIPS, 2019. PowerSGD uses power iteration to decompose the original gradient matrix $M$ into two $r$-rank matrices $P$ and $R$. This work is relevant and similar to the present work and requires mentioning.

2. In the expected loss minimization problem, what is $m$? Is it the number of observations in the training set? I did not find it before.

3. I have questions about solving the problem (10). How in (10), the authors arrive at quantized, $C_q, Z_q$? By using gradient descent, one can reach the factors, $(C, Z)$, and then it requires further projection by using quantization operations (as defined in (7)) to find quantized, $C_q, Z_q$. Running gradient descent algorithm directly on (10) does not guarantee quantized, $C_q, Z_q$. In the present form, it is misleading, and the treatment and evolution of the problem throughout the draft are highly faulty.

4. **Severely faulty mathematics.** Proposition A.1 is not correct. Please allow me to explain. The projection is not on a convex set; projection onto the set of sparse matrices (by using hard threshold operation) is a nonconvex projection. Moreover, the function that the authors defined in (10) is the same as the function, $f(x)$ defined in (14), which is a nonconvex function. First of all, for nonconvex functions proving convergence in terms of the optimum is impossible because one cannot guarantee the existence of a global minimizer $x_*$. Although the authors applied the projected gradient descent technique to nonconvex optimization, the convergence proof techniques used in the convex case do not apply directly to nonconvex problems. Therefore, the projection claim that the authors provided is mathematically incorrect. Please note that the claim that the authors made about the stationary point is erroneous. Moreover, the convergence rate with the derivation of the L-Lipschitz to justify their faulty analysis in Proposition A.1, is severely flawed. For (local linear) convergence of gradient descent with nonconvex projection, please see [1,2,3].

5. SVD stands for Singular Value Decomposition. Therefore, “SVD decomposition” does not make any sense.

[1] Lewis et al. Local linear convergence for alternating and averaged nonconvex projections. Foundations of Computational Mathematics, 9(4):485–513, 2009.
[2] Lewis and Malick. Alternating projections on manifolds. Mathematics of Operations Research, 33(1):216–234, 2008.
[3] Dutta et al., A Nonconvex Projection Method for Robust PCA, AAAI, 2019.





**Summary Of The Paper:**

The authors proposed a matrix factorization-based method for deep neural network weight compression. The weight tensors are factorized as two low precision quantized matrices, out of which one is sparse.

**Summary Of The Review:**

I did not read the paper after Section 3.1.1. The article is erroneous and the results are misleading. Please refrain from misleading the readers. This alone makes me recommend a STRONG REJECTION of the paper.

---

> ### Author Response · Authors · 2021-11-13
> **Answer**
>
> We thank the reviewer for the comments and we address the concerns below.
> 1. Thank you for mentioning the paper by Vogels et. al. This method uses dense factorization in the context of gradients compression, while quantization and sparsity are not used for further size reduction. We added the reference to related work.
> 2. Thank you for mentioning this. Indeed, we only use a limited subset of training data for optimization, and $m$ is the size of the subset. We clarified this in the paper.
> 3. While solving the problem (10), we obtain the quantized values of $C$ and $Z$ as outputs of $Q_c$ and $Q_z$, so they do not require further projection. While the values of $C$ and $Z$ are not quantized, we use the straight-through estimator in order to backpropagate through the quantization operation. This allows us to compute the updates for $C$ and $Z$.
> 4. Regarding severely faulty mathematics. First of all, as we mentioned in the paper,
> the theoretical support holds only  "for well behaved optimization problems" (see above 3.1.2). The goal of Appendix A is to provide intuition behind the choice of iterative projection methods and not proving convergence results for it.
> Regarding Proposition A.1 and the reviewer's comment, if we understand the comments correctly, we do not assume projection onto a convex set, and this is not mentioned anywhere in the paper.
> Besides, we never mentioned that the theory applies to the problem (10). It only provides an intuition for the chosen algorithm.
> The claim regarding stationary point is indeed correct.
> If $\mathbf{x_\star}$ is in $\mathcal{D}$ with  $\Delta_f(\mathbf{x_\star})=0$,  then we have
>
>     $\Pi_{\mathcal{D}} (\mathbf{x}_\star-\Delta_f(\mathbf{x_\star}))=$
>
>     $=\Pi_{\mathcal{D}} (\mathbf{x}_\star)=\mathbf{x}_\star$
> which means that $\mathbf{x}_\star$ is the stationary point of the projection step.
> Note that we are not providing general convergence results for general non-convex optimization. It is just shown that only under certain conditions on the projection step, we can get convergence results. We do not claim anywhere in the paper that these conditions hold for our problem. The same holds for $L$-Lipschitz analysis.
> To summarize, the theoretical statements of Appendix A are fully correct, and their goal is just to provide intuition about the algorithm 1 and not to prove any result about it.
>
> We hope this clarified your concerns and hope you are willing to review the full paper.

---

> > ### Comment · Reviewer_rRp9 · 2021-11-17
> > **Reply**
> >
> > The authors do not need to provide intuition about the theoretical guarantee of projected gradient descent while the function is convex and projection is performed on a convex set---simply for "well-behaved" problems. These are well-known results. What the authors are solving is a nonconvex problem (objective function) with nonconvex projection (for the sake of argument, I am not even considering the quantization steps, the hard thresholding alone is a nonconvex projection on the set of sparse matrices) and NO THEORETICAL GUARANTEE of projected gradient descent on convex set with "well behaved" problems holds there. This is a faulty and vastly misleading mathematical claim, and I will keep my score intact just for that.
> >
> > New questions:
> >
> > The authors said in reply to my comment: "we use the straight-through estimator in order to backpropagate through the quantization operation." What is that? Can the authors please explain further?
> >
> > I request the authors to check Algorithm 1. If what the authors are claiming is correct then step 5 of Algorithm 1 is not correct in its present form. What Step 5 of Algorithm 1 is returning is C, Z and they are not the same as quantized C, Z as the authors defined in equations 5 and 6, respectively. The pseudocode in Algorithm 1 is not in conformity with the process defined.
> >
> > In summary, there is a lot of mathematical jargon, and most of them are incorrect and misleading.

---

> > > ### Author Response · Authors · 2021-11-19
> > > **Answer**
> > >
> > > Thank you for your comments. As we mentioned, we never claim that the problem we are solving is convex. While the purpose of Proposition A.1. is to describe the conditions on the objective which would guarantee the convergence despite we perform a projection on a non-convex set. Again, we do not claim this is applicable to the problem we are solving, the purpose of adding this appendix to the paper is rather to provide intuition on convergence in some settings with a non-convex projection.
> > >
> > > Straight-through estimator (STE) is a technique used for back-propagation through the rounding operation which is non-differentiable. The STE gradient for the rounding operation is defined as $\frac{\partial \lceil p\rfloor}{\partial p}=1$, we added this to the paper in section 3.1.1. STE was introduced by Bengio et al. 2013 and is commonly used for training quantized models (Krishnamoorthi et al. 2018, Esser et al. 2019, Bhalgat et al. 2020, Fan et al. 2020).
> > >
> > > Step 5 of algorithm 1 is returning both updated $C, Z$ and their quantized versions $C_q, Z_q$ which are computed using equations 5 and 6. The computation of the updated $C, Z$ involves back-propagation through equations 5 and 6, so the computation includes the forward pass which also computes $C_q, Z_q$. The description of Algorithm 1 is fully correct.
> > >
> > > We hope our comments help to clarify the paper.

---

### Decision · Program_Chairs · 2022-01-20

**Decision:**

Reject

**Comment:**

Reviewer rRp9 expressed concerns regarding the theoretical results included in Appendix A. In the discussion (not visible to the authors), the AC and Reviewer zn4a agree that the exposition in the original manuscript was confusing and could lead readers to assume these results were valid for the proposed algorithm. Also, in the original manuscript the presentation of the theoretical results in the appendix was quite poor (e.g. Proposition A.1). Having said that, the contributions and main points of the work are not affected by these observations as it is mainly an empirical study.

Following from the previous point, Reviewers rRp9 and zn4a pointed out that the overall presentation of the method, particularly the mathematical presentation could be improved.

Reviewer zn4a points out that the method is not particularly novel, this was also indicated as a weakness by Reviewer iyVU. The main contributions of the work are to simultaneously solve the tensor factorization and vector quantization problems usinga form of projected gradient descent (with hard-thresholding). While the empirical results seem promising, are somewhat limited. The authors could make them stronger by studying other applications on top of image classification (e.g. semi-supervised setting, object detection or segmentation).

In the discussion (not visible to the authors), Reviewer iyVU stated in light of the other reviews, he/she does not oppose rejecting the work.

Overall, the method is technically sound and produces promising results. In its current form, however, the paper is not yet ready for publication. The AC encourages the authors to incorporate the feedback and resubmit the work to a different venue.